# Tailoring protein nanomechanics with chemical reactivity

Amy E.M. Beedle[1], Marc Mora[1], Steven Lynham[2], Guillaume Stirnemann[3] & Sergi Garcia-Manyes[1]

The nanomechanical properties of elastomeric proteins determine the elasticity of a variety of tissues. A widespread natural tactic to regulate protein extensibility lies in the presence of covalent disulfide bonds, which significantly enhance protein stiffness. The prevalent *in vivo* strategy to form disulfide bonds requires the presence of dedicated enzymes. Here we propose an alternative chemical route to promote non-enzymatic oxidative protein folding via disulfide isomerization based on naturally occurring small molecules. Using single-molecule force-clamp spectroscopy, supported by DFT calculations and mass spectrometry measurements, we demonstrate that subtle changes in the chemical structure of a transient mixed-disulfide intermediate adduct between a protein cysteine and an attacking low molecular-weight thiol have a dramatic effect on the protein's mechanical stability. This approach provides a general tool to rationalize the dynamics of S-thiolation and its role in modulating protein nanomechanics, offering molecular insights on how chemical reactivity regulates protein elasticity.

[1] Department of Physics and Randall Division of Cell and Molecular Biophysics, King's College London, WC2R 2LS London, UK. [2] Centre of Excellence for Mass Spectrometry, King's College London, SE5 8AF London, UK. [3] CNRS Laboratoire de Biochimie Théorique, Institut de Biologie Physico-Chimique, Univ. Paris Denis Diderot, Sorbonne Paris Cité, PSL Research University, 13 rue Pierre et Marie Curie, 75005 Paris, France. Correspondence and requests for materials should be addressed to S.G.-M. (email: sergi.garcia-manyes@kcl.ac.uk).

Modular proteins represent a natural strategy to achieve highly elastic materials with enhanced mechanical properties[1]. For example, the giant protein titin is formed by a large number of stiff immunoglobulin (Ig) and Fibronectin-like (Fn) domains intercalated between non-structured, extensible sequences (PEVK, N2B) that are mechanically compliant[2–4]. Combined, the distinct mechanical properties of both structurally diverse components regulate the large-scale passive elasticity of muscle under physiological conditions[2]. At the local scale, modulation of the nanomechanics of a single protein is mainly achieved through four specific molecular mechanisms; first and foremost (i), forced unfolding converts the mechanically rigid native state into an elastic unfolded conformation devoid of mechanical stability[5]. (ii) Secondly, a simple strategy to alter the mechanical properties of the protein's native state is through site-directed mutagenesis in well-defined positions within the force-bearing ('mechanical clamp') structural motif[6]. Alternatively, (iii) ligand binding can drastically increase the mechanical stability of a folded protein[7–9], or prevent an unfolded polypeptide from recovering its native mechanical stability by blocking successful refolding[10]. Both strategies often result in an all-or-none switch between both extreme mechanical stabilities[11]. (iv) A far less explored mechanism to modulate protein mechanics is achieved via post-translational modifications, through subtle yet crucial changes in chemical reactivity within the protein core[12,13]. The most relevant protein modification with a mechanical impact is arguably the formation of disulfide bonds, which are crucial modulators of protein extensibility. Disulfide bonds establish a rigid, molecular shortcut that impairs the complete force-induced unfolding of a variety of proteins including titin[14], and the extracellular cell adhesion molecules (CAM) protein superfamily[15]. In these cases, modulation of protein elasticity is generally dynamic and occurs under redox control[16]; while oxidizing conditions guarantee the presence of the covalently rigid disulfide bond, reducing conditions induce its rupture, thereby triggering full protein unfolding under force. Hence, the ability of the protein to form a disulfide bond during its folding route, a process known as oxidative folding[17], emerges as a crucial functional determinant of protein mechanics.

*In vivo*, the formation of disulfide bonds occurs through two independent pathways[18]. Within the 'redox route', a cysteine thiol can be oxidized via one or two electrons to a higher oxidation state species. Most typically, the free thiol is first oxidized to sulfenic acid, which then condenses with a neighbour protein thiol to form a protective disulfide bond that prevents irreversible protein damage due to cysteine overoxidation[13,19]. The second and most prevalent means of disulfide bond formation is catalysed by a superfamily of dedicated oxidoreductases—including thioredoxin, glutaredoxin, the bacterial Dsb and the eukaryotic protein disulfide isomerase enzymes—that share a common active site containing two highly conserved catalytic cysteines[20,21]. Their mechanism of action employs a thiol-exchange nucleophilic reaction, creating a transient yet obligatory mixed-disulfide intermediate between a substrate cysteine and the *N*-terminal cysteine of the enzyme's active site[22]. The determinant role of the mixed disulfide conformation to orchestrate oxidative folding has deserved great attention[23,24]. While the static conformational picture has been accessed through a variety of protein structures[25–29], the dynamics of this crucial fleeting mixed disulfide intermediate has only now begun to be understood[30,31].

Inspired by the functioning of oxidoreductases enzymes, we now aim to acquire a deep understanding of the naturally employed chemical tactics to rationalize the molecular mechanisms underpinning other closely related protein S-modifications.

We will use this *ad hoc* acquired knowledge to design new experimental strategies that rationally exploit the chemical properties of naturally occurring small molecules to modulate protein elasticity. A main advantage of this experimental approach is that it is not restricted to enzymatic activity, thus having the potential to be scaled up to large quantities, it has a predictive character and it offers exquisite control over the chemical selectivity of the small molecules as it avoids the inextricable conformational changes involved during protein-protein interactions that define enzymatic catalysis.

The most obvious candidates for the small chemical modulators are low-weight molecular thiols (LMW-SH)[32], especially encompassing cysteine (Cys), cysteinylglycine (CysGly), homocysteine (Hcys) and glutathione (GSH). Present both intracellularly and also in the human plasma, these small thiols are often regarded as biomarkers for oxidative stress[33], and their presence as mixed-disulfide with proteins increases with age[34,35]. While GSH reactivity has been the focus of extensive research[36,37], the reactivity of Cys and Hcys and their related protein post-translational modifications, namely S-cysteinylation and S-homocysteinylation, have comparatively received much less attention. In particular, high levels of Hcys have been associated to reduced muscle function and integrity[38]. In fact, hyperhomocysteinemia is considered as an independent risk factor for cardiovascular disease[39–42], and has been linked to pathological alterations in mechanically functional proteins such as the impaired fibrillin–fibronectin assembly[43]. Despite important efforts, a direct molecular link between the presence of S-homocysteinylation and the pathological mechanical effect is not completely understood.

Here we use single-molecule force-clamp spectroscopy, supported by mass spectrometry (MS) and density functional theory (DFT) calculations, to provide mechanistic insights into the direct link between the chemical reactivity of biologically relevant LMW thiols and their effect on mechanical protein folding. Our results demonstrate that the life-time of the ephemeral mixed disulfide intermediate structure is an essential modulator of the nanomechanics of cardiac titin, switching between two successfully refolded conformations with markedly different stiffness properties.

## Results

**S-cysteinylation promotes non-enzymatic oxidative folding.** We applied a constant force to a polyprotein composed of eight identical domains of a titin Ig27 mutant (I27$_{E24C–K55C}$)$_8$ that harbours a buried disulfide bond between amino acids 24 and 55 (Fig. 1a). A specifically designed five-pulse force protocol measures the protein elongation over time in the presence of 13.3 mM of L-cysteine (Fig. 1b), enabling the independent capture of each individual disulfide bond rupture and reformation event, with single bond resolution (Fig. 1c)[13]. Applying a short (500 ms) pulse of 150 pN to the (I27$_{E24C–K55C}$)$_8$ polyprotein elicits steps of ~15 nm, corresponding to the unfolding of the protein up to the position of the rigid disulfide bond (grey), which becomes exposed to the solution. A second pulse at a higher force (300 pN) triggers a ~10 nm stepwise increase in the protein length (red), corresponding to the force-catalysed rupture of each individual disulfide bond by the attacking L-cysteine nucleophile. The initial pulse, composed of ~10 nm and ~15 nm steps (Fig. 1d), results in a fully stretched and reduced protein form exhibiting a free cysteine thiolate and a mixed disulfide conformation between the native cysteine and the attacking L-cysteine (Fig. 1b).

Removal of the pulling force for $t_q = 8$ s (a period of time that is long enough to ensure quantitative protein refolding) triggered the collapse and folding of the protein. A second test pulse that

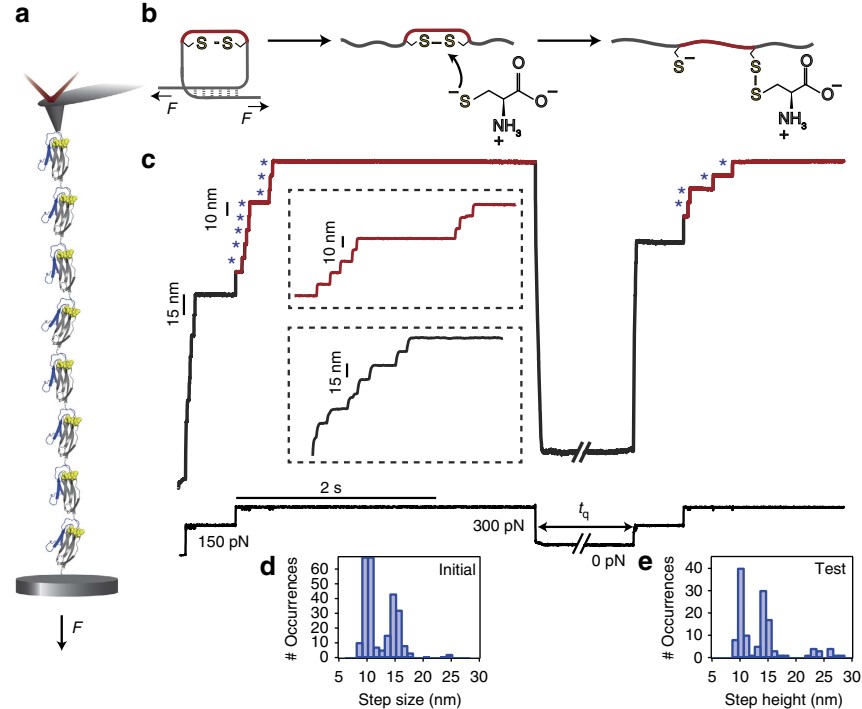

**Figure 1 | L-cysteine mediates the non-enzymatic reformation of disulfide bonds.** (**a**) A polyprotein construct formed of eight identical repeats of a mutated I27th Ig domain of titin containing a buried disulfide bond (I27$_{E24C-K55C}$)$_8$ is tethered between a gold substrate and an AFM cantilever tip, which applied a constant stretching force throughout the experiment. (**b**) Upon the application of force, the protein unfolds (grey) and exposes the disulfide bond to a L-cysteine solution (13.3 mM, pH = 7.2), which is able to attack the disulfide bond through a $S_N2$ nucleophilic attack, marked by the extension of the amino acids that were trapped behind the disulfide bond (red). The force-induced cleavage of the protein disulfide results in a stretched protein conformation containing a reduced protein thiol and a mixed disulfide bond between the attacking L-cysteine and a native cysteine. (**c**) Experimental folding trajectory of the (I27$_{E24C-K55C}$)$_8$ revealed by a five-pulse force quench protocol. The initial pulse unfolds the protein and ruptures the disulfide bond; the first pulse $F = 150$ pN unfolds and extends all the modules of the (I27$_{E24C-K55C}$)$_8$ in steps of ∼15 nm (grey inset), exposing the disulfide bonds to the L-cysteine solution. The second pulse at $F = 300$ pN catalyses the rupture of each disulfide bond (blue asterisks), giving rise to a staircase-like increase of protein length occurring in ∼10 nm steps (red inset). Quenching the pulling force ($F = 0$ pN) triggers the protein to collapse and fold for a time quench $t_q = 8$ s. The subsequent test pulse probes the folding status of the protein. Mirroring the initial pulse, a force pulse of $F = 150$ pN unfolds and extends the refolded protein up to the newly created disulfide bond (grey line) whereas the last $F = 300$ pN pulse is able to rupture it again (red line). The step-size histograms corresponding to the (**d**) initial and (**e**) test pulse highlight in both cases the presence of ∼10 nm and ∼15 nm steps.

mirrors the initial pulse probes the folding success of the folding reaction for a given $t_q$. Similar to the initial pulse, the elongation of the protein in the test pulse occurs in steps of ∼15 nm (unfolding) followed by the sequential reduction of each individual disulfide bond (∼10 nm steps, Fig. 1e). This sequence of events fingerprints the success of non-enzymatic oxidative folding process. In particular, the presence of ∼10 nm steps in the test pulse unambiguously hallmarks the non-enzymatic reformation of individual disulfide bonds, surprisingly occurring in the absence of any oxidasing agent, and its intricate link to the successful refolding of the protein.

**S-homocysteinylation blocks oxidative folding.** To probe the generality of the small thiol-mediated non-enzymatic oxidative folding process, we next probed the outcome of oxidative folding in the presence of the chemically analogous homocysteine nucleophile (Fig. 2a), only differing from L-cysteine in the presence of an extra methylene group next to the nucleophilic thiolate (Fig. 2b). Surprisingly, in this case the test pulse was mostly populated by steps of ∼25 nm, and the proportion of ∼10 nm steps, hallmark of disulfide reformation, was vanishingly small (Fig. 2a). The presence of ∼25 nm in the 150 pN test pulse stands for a mechanically weaker protein that has properly folded but has not reformed its internal stiff disulfide bond. The percentage of reduced, folded protein (∼25 nm steps) is dramatically

higher (∼40%) than in the case of L-cysteine (∼3%), whereas the situation is completely reversed in the case of disulfide bond reformation, which is favoured with L-cysteine (∼30%) and significantly reduced (∼10%) when Hcys is the attacking nucleophile (Fig. 2c,d). Altogether, these results conclude that the attacking nucleophile has a profound effect on the nano-mechanics of the refolded protein; while L-cysteine promotes a stiff, refolded protein with a properly reformed disulfide bond, S-homocysteinylation blocks oxidative folding but eventually allows the protein to fold into a more compliant native state conformation void of the reformed disulfide bond. The dramatic contrast in the effect that both analogous post-translational modifications have on the oxidative folding fate of the protein and on its nanomechanical properties is certainly surprising given the closely related chemical structures of both low-molecular-weight thiols.

**The mixed disulfide plays a key role in disulfide formation.** It is very plausible that the mechanism underpinning the non-enzymatic reformation of a protein disulfide bond observed for L-cysteine and also—to a much lower extent—for homocysteine, involves the reactivity of the mixed disulfide intermediate. MS measurements on the I27$_{E24C-K55C}$ monomer detected with high evidence the post-translational modification of both 24 and 55 cysteines by both L-cysteine and homocysteine (Supplementary

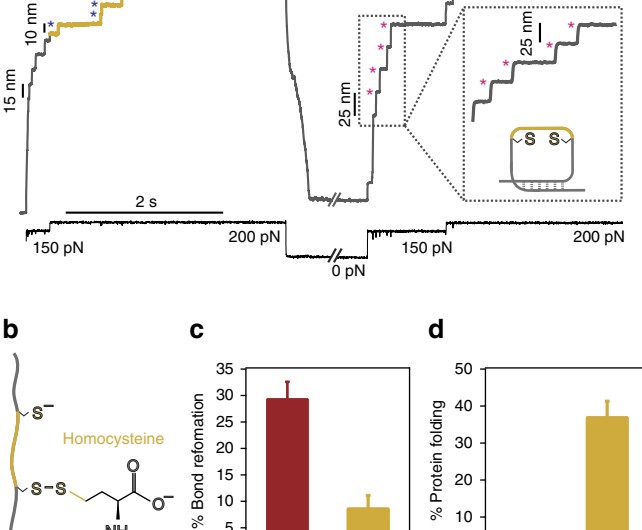

**Figure 2 | DL-homocysteine prevents successful non-enzymatic oxidative folding.** (**a**) The oxidative folding trajectory in the presence of homocysteine (111 mM, pH = 7.6) exhibits a test pulse that is composed of steps of ∼25 nm (marking the refolding of a reduced protein form, pink asterisks) followed by a vanishingly small population of ∼10 nm steps, fingerprinting disulfide bond reformation (blue asterisks). (**b**) The asymmetric mixed disulfide between a native cysteine and the attacking homocysteine, which only differs from L-cysteine in the presence of a methylene group (yellow) next to the nucleophilic thiol plays a crucial role in the outcome of the folding reaction. (**c**) While the percentage of disulfide bond reformation in the presence of homocysteine (yellow) is negligible compared to that of L-cysteine (red), (**d**) the situation is reversed for the population of folded and reduced proteins (∼25 nm steps), which is largely predominant for homocysteine (yellow) when compared to the residual fraction measured for L-cysteine (red) (error bar ± s.d.).

Figs 1,2 and Supplementary Table 1), which unambiguously confirmed the presence of the mixed disulfide. Noteworthy, given the different degree of solvent exposure of the protein disulfide bond obtained in MS and mechanical measurements, the yield of protein modification is not comparable in both experimental approaches. To prove the necessary participation of the mixed disulfide adduct in the dynamics of disulfide bond reoxidation, we conducted analogous experiments on an equivalent (I27$_{G32C-A75C}$)$_8$ polyprotein using the phosphorus-based TCEP nucleophile (we chose the I27$_{G32C-A75C}$ mutant, harbouring the buried disulfide bond in a different position within the protein structure, because TCEP can partially reduce the disulfide bond of the (I27$_{E24C-K55C}$)$_8$ polyprotein in the absence of force[44]). In the case of the P-based nucleophile TCEP, the force-induced S$_N$2 disulfide cleavage[45] obviously does not yield a stable mixed disulfide, but rather a highly reactive mixed S–P thioalkoxyphosphonium cation intermediate[46] (Supplementary Fig. 3), which can be easily displaced by water molecules to render a completely unfolded protein with two reduced thiolates (Fig. 3a). Analysis of the test pulse in our refolding trajectories (Fig. 3b) reveals the absolute absence of ∼10 nm steps, indicating that no disulfide bond is present in the otherwise successfully refolded protein (confirmed by the presence of ∼25 nm steps, Fig. 3c). These experiments demonstrated the crucial implication of the mixed disulfide species in the thiol-mediated mechanism of oxidative folding.

**Homocysteine exhibits an anomalous high p$K_a$.** To elucidate the molecular origin of the markedly different behaviour observed for L-cysteine and homocysteine, we investigated how the physico-chemical properties of both small thiols could determine their reactivity within the context of a nucleophilic substitution (S$_N$2) reaction. It is well established that a charged thiolate will be a better nucleophile than a protonated thiol[47]. Hence, there is a general direct connection between the goodness of a particular nucleophile/leaving group and its acid–base properties. Although basicity is a thermodynamic property and nucleophilicity is a kinetic phenomenon, a good correlation between both properties has been observed, generally following the rule of thumb stating that good leaving groups are the conjugate bases of strong acids[48]. While L-cysteine is reported to exhibit a p$K_a$ = 8.15 (ref. 49) − 8.3 (ref. 50), there is a surprising disagreement in the literature regarding the p$K_a$ value for homocysteine, for which two largely disparate values (p$K_a$ = 8.7 (ref. 51) and p$K_a$ = 10 (refs 50,52,53)) are reported. To resolve such a dichotomy, we independently measured the p$K_a$ of both compounds by monitoring the free thiol absorbance at $\lambda$ = 240 nm (refs 54,55) as a function of pH for both L-cysteine and homocysteine compounds (Fig. 4a). While an expected p$K_a$ = 8.4 was measured for L-cysteine, the p$K_a$ of homocysteine was found to be significantly higher (p$K_a$ = 9.9) in our experimental conditions, supporting previous findings[50]. Crucially, our homocysteine measurements required reducing conditions to attenuate its fast dimerization, especially under basic conditions (Supplementary Fig. 4). The dissimilar p$K_a$ value found for both isolated compounds in solution is likely to account for their different reactivity when selectively bound to a protein via a protein post-translational modification.

**DFT calculations provide a thermodynamic perspective.** To obtain a more precise understanding of the thermodynamically allowed reactivity between the different thiol species involved in our experiments (namely (i) the initial disulfide bond between two structurally close cysteines, (ii) the newly mixed disulfide bond species; (iii) the resulting free cysteine thiol after disulfide bond cleavage and (iv) the free small thiol nucleophile species in solution), we performed DFT calculations on model thiol and disulfide species using the expanded 6-311 + G(d,p) basis set and the B3LYP functional, which has been used in the context of thiol-disulfide exchange[56] (Table 1). Noteworthy, in this simplified approach any free-energy contributions arising from a protein conformational change upon formation of a native or mixed disulfide are neglected (only the free-energy associated with an isolated disulfide bond rupture is computed).

The attack of the Cys$_{prot}$–Cys mixed disulfide bond by a solution L-cysteine is, from a free-energy perspective, an equilibrium process with no net energy contribution ($\Delta G$ = 0 kcal mol$^{-1}$ reaction 1, Table 1), whereas the rupture of the mixed non-equivalent disulfide between Cys$_{prot}$–Hcys by free Hcys in solution (reaction 2, Table 1) is favourable ($\Delta G$ = − 4 kcal mol$^{-1}$). By contrast, the attack of the same mixed disulfide (Cys$_{prot}$–Hcys) by the freed protein cysteine (reaction 3, Table 1) is not favourable ($\Delta G$ = + 2.6 kcal mol$^{-1}$). Hence, these results can rationalize a few key non-expected experimental observations, namely that: (i) the symmetric mixed disulfide formed by two equivalent cysteines can be attacked by the freed protein thiolate when in close physical proximity to reform a disulfide bond (Fig. 4b); by contrast, (ii) the asymmetric Cys$_{prot}$–Hcys is not quantitatively attacked by the free thiol protein cysteine to reform the previously cleaved disulfide bond but (iii) can be attacked by free homocysteine molecules in solution, thus resolving the mixed disulfide and leaving the protein with two reduced thiolates able to successfully refold in the absence of a formed disulfide bond giving rise to the ∼25 nm steps.

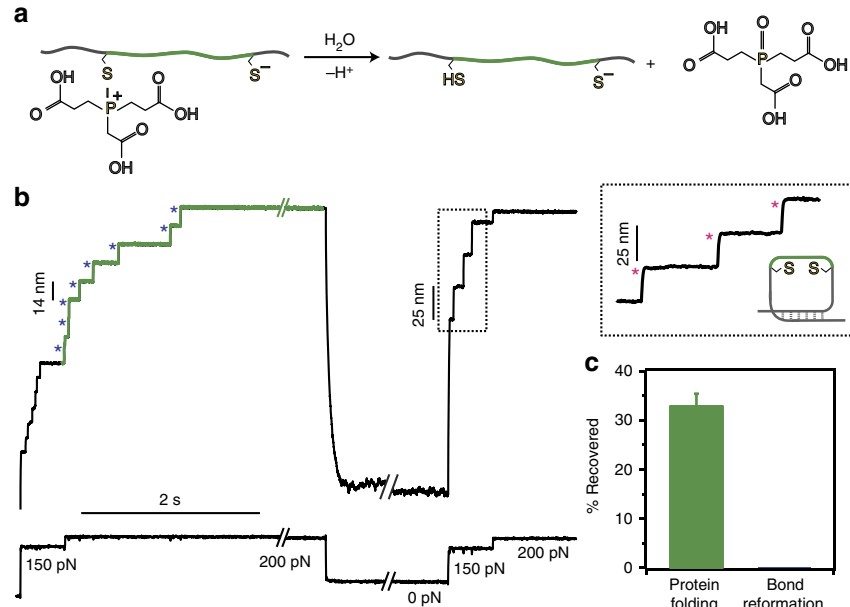

**Figure 3 | TCEP does not create a mixed disulfide and therefore the protein refolds into a conformation void of the disulfide bond.** (**a**) When a phosphorous nucleophile such as TCEP is used to induce the force-activated cleavage of the disulfide bond on a structurally analogous (I27$_{G32C-A75C}$)$_8$ polyprotein (4 mM, pH 7.2), a highly unstable mixed S–P thioalkoxyphosphonium cation intermediate is formed, which can be rapidly resolved by water solvolysis. TCEP-mediated disulfide bond rupture results in a stretched protein harbouring two reduced cysteines. (**b**) The test pulse in the folding trajectories is composed of steps of ∼25 nm (pink asterisks), fingerprinting a natively folded protein devoid of the rigid disulfide bond. (**c**) Crucially, the absence of a mixed disulfide precludes the reformation of the native disulfide bond in the context of non-enzymatic oxidative folding (error bar ± s.d.).

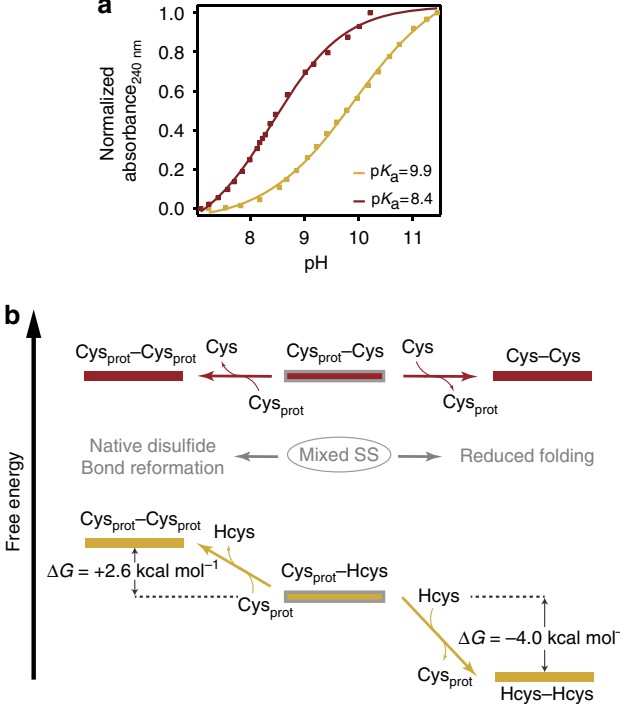

**Figure 4 | The high p$K_a$ of homocysteine leads to a thermodynamically stable asymmetric mixed-disulfide.** (**a**) Free homocysteine exhibits a surprisingly higher p$K_a$ = 9.9 (red) than the homologous L-cysteine (p$K_a$ = 8.4, yellow) revealed by absorbance measurement at $\lambda$ = 240 nm in solution. (**b**) DFT calculations predict the high stability of the Cys–Hcys mixed disulfide, which cannot be attacked by a neighbouring protein free thiol ($\Delta G$ = + 2.6 kcal mol$^{-1}$) to reform the native disulfide.

**Table 1 | DFT free-energy calculations corresponding to the disulfide-exchange reactions induced by the studied low-molecular-weight thiols.**

| | Reaction | $\Delta G$ (kcal mol$^{-1}$) |
|---|---|---|
| 1 | Cys-Cys + Cys$^-$ → Cys-Cys + Cys$^-$ | 0.0 |
| 2 | Cys-Hcys + Hcys$^-$ → Hcys-Hcys + Cys$^-$ | − 4.0 |
| 3 | Cys-Hcys + Cys$^-$ → Cys-Cys + Hcys$^-$ | + 2.6 |
| 4 | Cys-NAC$^-$ + Cys$^-$ → Cys-Cys + NAC$^{2-}$ | + 7.0 |
| 5 | Cys-NAC$^-$ + NAC$^{2-}$ → NAC-NAC$^{2-}$ + Cys$^-$ | − 7.7 |

**The distinct nucleophilic power of L-cys and Hcys**. While free-energy DFT calculations agree with our (thermodynamic) acid–base measurements, full understanding of the S$_N$2 nucleophilic substitution reaction would also require kinetic considerations. At the fundamental level, thiol-disulfide exchange is a classical bimolecular nucleophilic exchange S$_N$2 reaction that occurs through a linear (S–S–S) Walden transition state complex[47,56,57]. Crucially, mechanical force accelerates disulfide cleavage by lowering the kinetic barrier[45,58,59]. To compare the nucleophilic power of both small thiols, we measured the rate of disulfide reduction at the same stretching force ($F$ = 200 pN) for two solutions containing the same concentration of active, deprotonated thiolate species at the same pH (Fig. 5a). Our results demonstrate that the rate of disulfide reduction is clearly higher for homocysteine ($r$ = 1.11 s$^{-1}$) than for cysteine ($r$ = 0.08 s$^{-1}$), thus highlighting the better nucleophilic nature of homocysteine (Fig. 5b). From the computational perspective, a very crude estimate of the relative nucleophilic character of both cysteine and homocysteine compounds can be obtained by comparing the energies of the HOMO orbitals of the isolated small compound. Our calculations revealed (Supplementary Fig. 5) that the HOMO energy for Hcys is higher than that of L-Cys, suggesting an enhanced reactivity toward electrophiles.

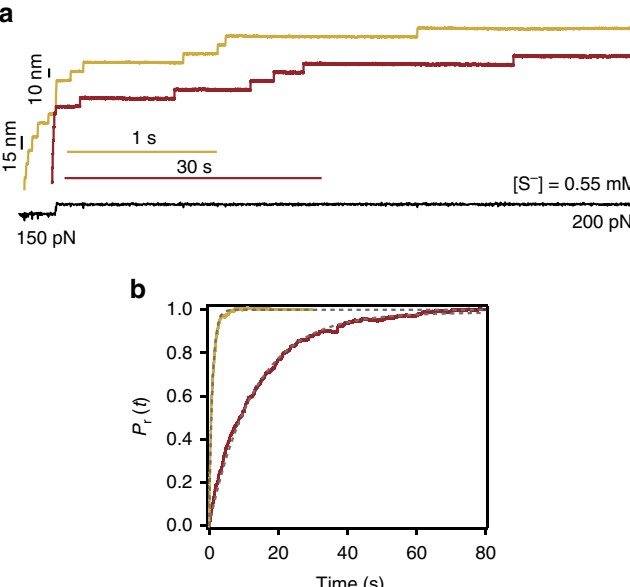

**Figure 5 | The kinetics of disulfide reduction is largely accelerated by homocysteine due to its high nucleophilic power.** (**a**) A two-pulse force protocol measures the kinetics of disulfide reduction when the protein is exposed to the same concentration of active deprotonated species of L-cysteine (red) and homocysteine (yellow). The time-course of disulfide bond reduction is dramatically longer for L-cysteine as illustrated by the drastically different scale bars (**b**) The rate of disulfide bond reduction at $F = 200$ pN is $\sim$14-fold faster for homocysteine ($r = 1.11\,\mathrm{s}^{-1}$) than for L-Cys ($r = 0.08\,\mathrm{s}^{-1}$) under the same experimental conditions, demonstrating that homocysteine is a better nucleophile than L-cysteine.

Similarly, the effective electrostatic negative charge localized on the attacking sulfur atom in homocysteine is more negative ($-0.81e$) than in L-cysteine ($-0.77e$). Altogether, these results suggest that Hcys is a better nucleophile (and thus, a worse leaving group) than L-Cys (Supplementary Fig. 5), qualitatively agreeing with the experimental results and the DFT free-energy calculations (Table 1).

Noteworthy, precise quantification of both thermodynamic and kinetic parameters defining the system would require taking into account the subtle yet important effects of the larger and complex protein environment, and also the non-trivial, catalytic effect that the pulling force has on the reaction kinetics[58,59]. Further free-energy calculations at the QM/MM level with explicit solvent, hopefully coupled with a force perturbation[60–62] (which goes beyond the scope of the current work), will help elucidate the finer parameters (for example, solvation, bond orientation[62]) regarding the mechanistic dynamics of the process. In any case, our simple estimates regarding thermodynamic and kinetic considerations of the isolated reactants provide a good general qualitative description of the experimental results, describing a localized chemical reaction occurring in the core of a much complex protein environment.

**The experimental approach predicts the reactivity of NAC.** To further test if these considerations are sufficient predictors of disulfide reactivity and thus, of the final outcome of the oxidative folding reaction, we conducted similar nanomechanical experiments using N-acetylcysteine (NAC) as a nucleophile (Fig. 6a,b). NAC is a drug derivative of cysteine with an acetyl group attached to its nitrogen (Fig. 6a), which has been used in clinical practice for decades as a mucolytic agent and also employed in the

treatment of numerous disorders including paracetamol intoxication and vascular and cardiac injury[63]. The uniqueness in the action of NAC stems from its particular physicochemical properties, which render it an optimal reducing agent of protein disulfide bonds[64]. Measurement of the $pK_a$ for NAC through our absorbance experiments yielded a value $pK_a = 9.6$ (Fig. 6c). Similarly, DFT calculations predicted that, from a thermodynamics perspective, the reaction whereby a free protein cysteine attacks a NAC–$Cys_{prot}$ mixed disulfide bond is energetically impaired ($\Delta G = +7\,\mathrm{kcal\,mol}^{-1}$, reaction 4, Table 1). By contrast, the NAC–$Cys_{prot}$ mixed disulfide can be attacked by free solution NAC molecules ($\Delta G = -7.7\,\mathrm{kcal\,mol}^{-1}$, reaction 5, Table 1). Moreover, HOMO and electrostatic calculations (Supplementary Fig. 5) highlight the high nucleophilic power of NAC with respect to L-cysteine. Hence, from the physicochemical viewpoint, NAC is closely related to Hcys. In effect, our single-molecule folding trajectories in the presence of NAC (Fig. 6b) certify that while the protein can fold into its reduced form (Fig. 6d), its ability to effectively reform the previously reduced disulfide bond is vanishingly small (showing an even smaller percentage than homocysteine, Fig. 6e). The presence of the NAC–protein mixed disulfide was further certified through MS analysis (Supplementary Fig. 6).

## Discussion

Detection of chemical intermediates is the Holy Grail in the elucidation of reaction mechanisms. However, due to their ephemeral nature, their capture is frequently challenging. An extra layer of complexity is added when the chemical reactions occur within the core of an individual protein—giving rise to the generally called post-translational modifications—mainly because the reactivity of the newly created intermediate adduct is intrinsically coupled with the conformational motions of the protein. Hence, direct observation of the chemical evolution of a protein-based chemical reaction and its crucial effect on the folding fate of a protein remains largely elusive.

Due to the highly polarizable sulfur atom, the amino acid cysteine is a particular target for numerous chemical modifications in the cellular milieu[65], of which the most commonly known is the formation of disulfide bonds. Such thiol-disulfide exchange reactions have crucial roles in biology, both stabilizing structural protein motifs and also as essential regulators of the dynamic functional properties of many enzymes, including most oxidoreductases[18]. Albeit perhaps less studied, another important natural example of disulfide bond formation involves small plasma thiols. Indeed, an increase in plasma disulfides, including low-molecular-mass protein-mixed disulfides, has been postulated as a biomarker of oxidative stress and aging. In particular, direct correlations were found between the disulfide forms of Cys and Hcys and age[33–35]. Several pathologies, including cognitive impairment, osteoporosis, eye's disorders, complications in pregnancy, reduced skeletal muscle function and integrity and cardiovascular disease have been directly correlated with high homocysteine levels[50]. Hence, it is enticing to speculate about the presence of a set of distinct chemical properties of Hcys that underlie its ability to dimerize and to form stable disulfide bonds with protein cysteine residues that impair or alter normal protein function. Several proteins with distinct biological function such as albumin[49], fibronectin, fibrillin, tropoelastin[43], transthyretin, alpha-crystallin[66] and metallothionein have been identified as natural Hcys targets[50]. While a few pioneering studies postulated a molecular-targeting hypothesis that link the S-homocysteinylation post-translational modification with protein function[50,52], direct experimental evidence remained elusive.

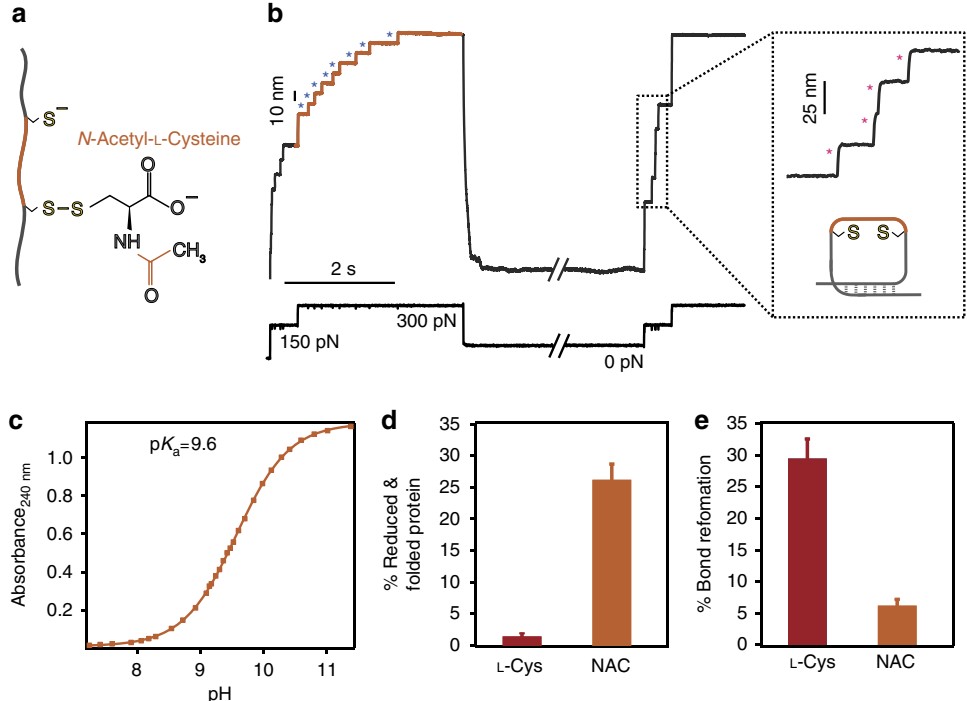

**Figure 6 | *N*-acetylcysteine creates a stable protein mixed-disulfide that largely blocks oxidative folding.** (**a**) The force-induced cleavage of a protein disulfide bond results in an asymmetrical mixed disulfide intermediate of a protein cysteine with NAC, a cysteine derivative with an acetyl group (orange) attached to its nitrogen. (**b**) The individual single-molecule trajectories track the success of the oxidative folding assay. The initial pulse results in the unfolding of the $(I27_{E24C-K55C})_8$ polyprotein (15 nm steps, grey) followed by the force-induced cleavage of the solvent-exposed disulfide bonds, marked by the 10 nm step-wise increase of the protein length (blue asterisks). By contrast, the test pulse is mostly composed of 25 nm steps (pink asterisks), which hallmark that the protein has successfully refolded into a reduced conformation that is void of the native disulfide bond. (**c**) $pK_a$ determination through absorbance measurements at $\lambda = 240$ nm yields a $pK_a = 9.6$. These results demonstrate that NAC is a weak acid, and hence its conjugated base will be a poor leaving group, in agreement with the high stability of the NAC-mediated mixed disulfide. (**d**) While the % of successful folding to a reduced conformation is largely prevalent in the presence of NAC, (**e**) the % of disulfide bond reformation is vanishingly small for NAC as compared to the structurally akin L-cysteine (error bar ± s.d.).

Our single-molecule approach enables us to independently monitor the protein folding and disulfide bond formation events, hence allowing direct observation of both the molecular mechanisms underlying oxidative folding and the effect that closely related post-translational modifications have on the protein's mechanical function. Here we present a chemical strategy to promote successful non-enzymatic oxidative protein folding that is based on the high reactivity of a symmetric $Cys_{prot}$–Cys mixed disulfide obtained through protein S-cysteinylation. In this scenario, being in energetic equilibrium with a neighbouring protein thiol, the shuffling between three equivalent cysteines when in close spatial proximity allows disulfide isomerization[67], eventually leading to the spontaneous reformation of the native disulfide bond. By contrast, homocysteine forms a highly stable asymmetrical disulfide bond with a protein cysteine that can hardly be re-attacked by the previously freed neighbour thiol. We hypothesize that the high propensity of Hcys to form stable mixed disulfides with protein cysteines can be a generic mechanism that underlies the lack of protein functionality upon protein S-homocysteinylation. Similarly, the high affinity of NAC (and in general, all the tested small thiols) to create disulfide bonds might explain its important role in protein protection under oxidative stress[68], avoiding irreversible thiol overoxidation to sulfinic and sulfonic acid.

A particular characteristic of our experiments is that chemical modifications occur on cryptic cysteines, which only become solvent exposed after mechanical unfolding of the protein. In this sense, the outcome of the folding process is not only limited by the chemical reactivity of the protein, but also by protein conformation, which is precisely modulated by mechanical force. Hence, our findings suggest a combined mechanochemical scheme that reveals how the distinct physicochemical properties and reactivity of closely related small thiols dictates the folding fate of the protein, giving rise to two successfully refolded forms, yet with very distinct mechanical properties (Fig. 7a). In our approach, the native disulfide bond is first reduced by the attacking nucleophile thiol upon application of force. While the redox potential is suggested to be slightly modulated by the pulling force[62], force-activated reduction of a protein disulfide bond is mostly under kinetic control[58], resulting in a protein stretched conformation containing a completely solvent-exposed mixed disulfide which is *a priori* susceptible to be attacked by the thiol nucleophiles present in solution.

Given that in the presence of TCEP (where no mixed-disulfide bond can occur) the protein successfully refolds (certified by the presence of 25 nm steps in the test pulse), we speculate that, in the presence of L-Cys, Hcys and NAC, the observed 25 nm steps (especially obvious with the experiments with Hcys and NAC) correspond to a scenario whereby free thiolates in solution are able to attack the mixed disulfide conformation (namely Cys–$Cys_{prot}$, Hcys–$Cys_{prot}$ and NAC–$Cys_{prot}$), therefore resulting in a protein containing two reduced thiolates that are able to properly refold, void of the disulfide bond (Fig. 7b). Indeed, DFT calculations show that the attack of solution free Hcys to the solvent-exposed, mixed asymmetrical Hcys–Cys mixed disulfide

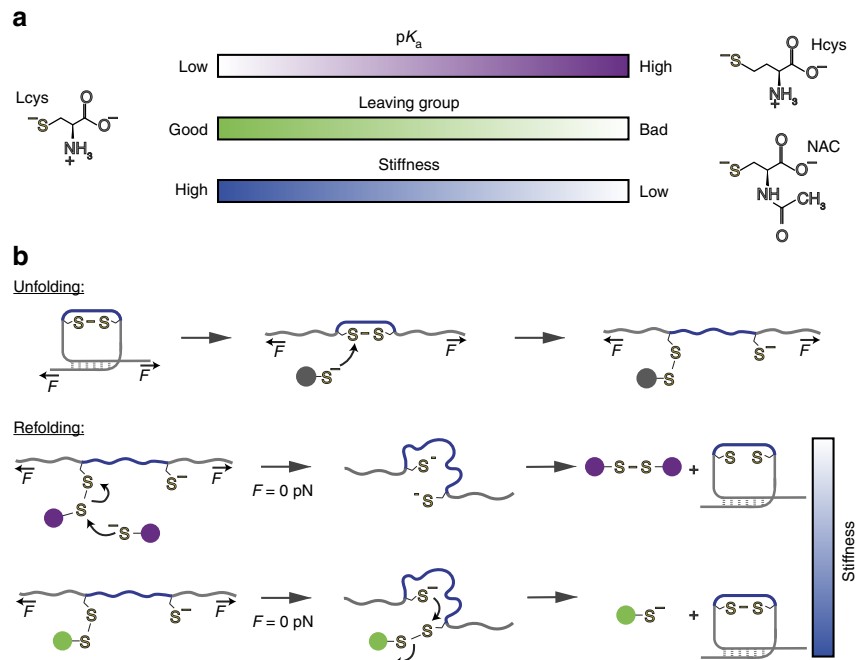

**Figure 7 | Protein reactivity induced by S-thiolation regulates protein nanomechanics. (a)** The physicochemical properties of the different small thiols govern the life-time of the mixed disulfide, which largely controls the outcome of the oxidative folding reaction and determines protein nanomechanics. (**b**) Schematics of the reactivity of the S-thiolated protein under force. On the application of mechanical force, the different small thiolates rupture a protein disulfide bond, creating a mixed disulfide and a reduced protein thiolate. In such extended and reduced protein conformation, the protein thiol that is bound to a strong nucleophile acts as the leaving group when attacked by free thiols in solution (purple spheres, for example, homocysteine and NAC) resulting in two reduced protein cysteines. In the absence of force ($F = 0$ pN), the reduced protein can recover a more compliant folded form that is devoid of the native disulfide bond. By contrast, when a softer nucleophile is used (green sphere, for example, L-cysteine), the highly reactive symmetric mixed disulfide is in equilibrium with the free protein cysteine thiol. In this scenario, disulfide bond isomerization occurs, resulting in a stiff folded protein characterized by the reformation of the native disulfide bond.

is thermodynamically very favoured ($\Delta G = -4.0$ kcal mol$^{-1}$, reaction 2, Table 1), and it is very likely that this equilibrium is further displaced due to the high insolubility of the Hcys–Hcys homodimer (additional small molecule MS experiments reveal that monomeric Hcys in solution undergoes oxygen-mediated oxidation to its dimeric, insoluble form[42] within the timescale of ∼1 h, Supplementary Fig. 7). Similarly, the attack of NAC free molecules in solution to the NAC–Cys$_{prot}$ heterodymer is even more favourable from the thermodynamic perspective ($\Delta G = -7.7$ kcal mol$^{-1}$, reaction 5, Table 1). These free-energy considerations are in very good agreement with our experimental observations, showing that while the percentage of reduced folding is vanishingly small for L-Cys (Fig. 2d), it is instead predominant for both Hcys (Fig. 2d) and also for NAC (Fig. 6d). Further experiments where the time the reduced protein is left extended and exposed to NAC molecules in the solvent (Supplementary Fig. 8) confirmed that the percentage of reduced folding (requiring a solution NAC molecule attacking the asymmetric disulfide bond) remained invariable over time, suggesting that the overall reactivity process seems to occur under thermodynamic control, probably reflecting the simplicity of the reaction and the lack of conformational and steric hindrance.

Supported by recent findings using a cysteine-defective thioredoxin mutant[30] and reduced glutathione[12], we speculate that the presence of the mixed disulfide blocks protein folding (giving rise to a compliant protein form), and that it needs to be resolved, either through the attack of a neighbouring protein cysteine to reform the native disulfide bond after protein collapse or via reaction with thiols free in solution. However, we cannot completely rule out at this stage the possibility that the protein is able to refold around its mixed-disulfide and regain mechanical stability.

Upon withdrawal of the applied force, the protein collapses into a molten globule conformation[69] where reactive cysteines are physically close and sterically buried from the protein environment. In the case of L-cysteine, disulfide bond isomerization[67] allows the reformation of the native disulfide bond, giving rise to a rigid protein form (Fig. 7b). By contrast, in the case of Hcys and NAC where the mixed disulfide bond has been resolved by reacting with free solution nucleophiles while the protein is stretched under force, quenching the force results in a refolded and reduced protein form that displays a mechanical stability that is intermediate between the mechanically stiff oxidized protein containing the disulfide bond, and the mechanically compliant misfolded protein that harbours a stable, non-resolved mixed disulfide species (Fig. 7b).

The measurements presented here are conducted on an Ig domain of cardiac titin, a protein that carries an important mechanical role *in vivo*. A key requirement of elastomeric proteins, always subject to multiple stretching–relaxation cycles, is their reversible functionality and reliability over time[70]. Our approach provides an extra level of understanding on the molecular mechanisms of disulfide bond reformation under force, and its direct link to mechanical folding. Crucially, our measurements reflect the largely distinct role of the closely related S-cysteinylation and S-homocysteinylation on protein elasticity, and add to the emerging research field that highlights the importance of force-induced exposure of cryptic sites to regulate protein nanomechanics, both via post-translational modifications, such as S-glutathionylation[12] and S-sulfenylation[13], or through the unravelling of metal binding sites[71]. More generally, our proof-of-principle approach demonstrates the use of highly localized chemical reactivity—which can be potentially

expanded to other distinct chemical mechanisms—to rationally tailor protein elasticity, and offer a new approach to engineer reversible redox-responsive protein-based biomaterials with adaptable mechanical properties.

## Methods

**Protein engineering.** The $(I27_{E24C-K55C})_8$ and $(I27_{G32C-A75C})_8$ polyproteins were constructed using the BamHI, BglII and KpnI restriction sites (polyprotein sequences shown in Supplementary Table 2). The engineered polyproteins and the $I27_{E24C-K55C}$ monomer (synthetic gene, Thermo Fisher Scientific) were then cloned into the pQE80L (Qiagen) expression vector, and transformed into the BLR(DE3) *Escherichia coli* expression strain (VWR International). Cells were grown at 37 °C and supplemented with $100 \,\mu g \,ml^{-1}$ ampicillin. After an $OD_{600}$ of ∼0.6 was reached, cells were induced with Isopropyl β-D-1-thiogalactopyranoside (1 mM) and incubated overnight at 20 °C. Cells were disrupted using a French Press. The polyproteins obtained after lysis were purified by metal affinity chromatography on Talon resin (Clontech) using equilibration buffer (50 mM sodium phosphate pH 7.0, 300 mM NaCl), washing buffer (50 mM sodium phosphate pH 7.0, 300 mM NaCl, 20 mM Imidazole) and elution buffer (50 mM sodium phosphate pH 7.0, 300 mM NaCl, 250 mM Imidazole). This was followed by gel-filtration using a Superdex 200 10/300 GL column (GE Biosciences) using PBS buffer pH 7.3. We measured the protein concentration using the Bradford assay.

**Force-clamp spectroscopy.** To perform the single-molecule force-clamp spectroscopy atomic force microscopy experiments we used both a home-made set-up[72] and a commercial Luigs and Neumann force spectrometer[73]. We deposited 1–10 μl of protein in PBS solution (at a concentration of 1–10 mg ml$^{-1}$) onto a freshly evaporated gold coverslide. Prior to each experiment we independently calibrated the cantilever ($Si_3N_4$ Bruker MLCT-AUHW) by means of the equipartition theorem, exhibiting a typical spring constant of ∼12–17 pN nm$^{-1}$. Individual polyprotein molecules were fished by pressing the cantilever onto the gold surface at forces spanning 500–1,500 pN, thus enhancing non-specific physiosorption. The deflection of the cantilever (force) was then kept constant throughout the experiment (up to ∼80 s) thanks to a proportional, integral and differential amplifier (PID) feedback system, achieving a time response to ∼2–5 ms. The data was filtered at 1 kHz with a pole Bessel filter. In the disulfide bond rupture kinetics experiments, the protein was first unfolded at 150 pN for 0.5 s, and subsequently the force was raised up to 200 pN and left constant for a large, variable period of time to capture the full kinetics of disulfide reduction for each studied nucleophile. In the oxidative folding experiments, the protein was unfolded at 150 pN for 0.5 s, followed by a high force pulse (varying between 200 and 300 pN depending on the use nucleophile) for 3 s. The quench pulse where the pulling force is completely removed was set in all cases to $t_q = 8$ s. Experiments were carried out by keeping the concentration of the deprotonated form of L-cysteine and N-aceyl-cysteine nucleophiles constant to 0.7 mM, whereas the deprotonated form of DL-homocysteine was kept at 0.55 mM due to solubility issues, and the total concentration of each nucleophile in the measuring solution was likewise adjusted according to the p$K_a$ of each compound. The different thiol nucleophiles (L-cysteine (Santa Cruz biotechnology, 98%), DL-homocysteine (Sigma-Aldrich, 95%), N-acteyl-cysteine (Sigma-Aldrich, 98%) and TCEP (Sigma-Aldrich)) were diluted into a sodium phosphate buffer solution, specifically, 50 mM sodium phosphate ($Na_2HPO_4$ and $NaH_2PO_4$), 150 mM NaCl. The final pH of in each solution was adjusted by adding the required amounts of HCl/NaOH solutions. Measuring solutions were filtered through a 0.2-μm membrane prior to each experiment. Each solution was freshly prepared prior to every experiment.

**Data analysis.** Both the acquisition and analysis of the experimental data was perfomed using customized software written in Igor Pro 6.0 (Wavemetrics). For the kinetics studies, we only analysed trajectories exhibiting at least five unfolding events (15 nm steps) followed by five reduction events (10 nm steps). We followed the same criteria for the oxidative folding experiments, and we considered only trajectories whereby the test pulse reached the full unfolded length characterizing the initial pulse. We used the boostrap method, to estimate the standard error of the mean (s.e.m.) for the refolding fraction.

**DFT calculations.** Electronic structure calculations using DFT were performed using the Gaussian 09 software[74]. The B3LYP correlation function[75,76] was employed with the 6-311 + G(d,p) basis set. All structures were optimized in implicit water solvent consisting of a dielectric medium using the PCM model[77]. Frequency calculations in the gas phase were performed on these optimized structures. The reported free energies correspond to the sum of gas-phase energy at 0 K, correction for solvation, zero-point energy, and thermal correction at 298 K. The HOMO energy levels, the electrostatic potentials and the charges (obtained using the Natural Bond Orbital analysis implemented in Gaussian) were determined from the optimized structures in implicit solvent.

To mimic protein–protein and mixed disulfide bond, we considered dimers of cysteines or its derivatives. At neutral pH, amino acids would be present in a zwitterionic $NH_3^+ \dots COO^-$ form. However, as already observed for amino acids

(including cysteine[78,79]), the neutral tautomer $NH_2 \dots COOH$ was found to be more stable in most cases. Only the inclusion of explicit water molecules would provide the necessary stabilization of the zwitterionic state by the formation of strong hydrogen bond with water[80], but it is beyond the scope of the current work. The reported energies therefore refer to the $NH_2 \dots COOH$ forms of thiolates for cysteine and homocysteine. For NAC, the $NH_2$ group is replaced by an amide group, which is not protonated at neutral pH, so the $NH \dots COO^-$ form was considered.

**Mass spectrometry experiments.** *Enzymatic digestion.* The $I27_{E24C-K55C}$ mono-mer samples were prepared by diluting 20 μl of protein ($0.5 \,\mu g \,\mu l^{-1}$) in 20 μl PBS pH 7.4 containing L-cysteine, homocysteine and N-acetyl cysteine at the same concentration as used in the single molecule experiments. Prior to the MS analysis, samples were exposed for 2 h at 37 °C to trypsin digestion at a ratio of 1:20 (enzyme:substrate).

*Liquid chromatography–mass spectrometry.* An EASY NanoLC system (ThermoFisherScientific, UK) was used to conduct the hromatographic separations. We used a reversed phase chromatography on a 75 μm C18 column to resolve the peptides from a total protein amount on column of 2 μg. A three-step linear gradient of acetonitrile in 0.1% formic acid was employed to elute the peptides at a flow rate of 300 nl min$^{-1}$ over 60 min. The eluate was ionized by electrospray ionization using an Orbitrap Velos Pro (ThermoFisherScientific, UK) operating under Xcalibur v2.2. Precursor ions were selected according to their intensity using the collision-induced fragmentation employing a Top20 CID method and a Top10 high-energy collision dissociation method. The MS/MS collision dissociation analyses were performed by using the collision energy profiles, which were chosen according to both the mass-to-charge ratio ($m/z$) and the charge state of the peptide.

**Ultraviolet spectrophotometric measurements for p$K_a$ determination.**
Absorbance measurements were performed with a JENWAY 6305 Spectro-photometer at room temperature. Absorbance measurements of L-cysteine and N-acetyl-L-cysteine at a final concentration of 250 μM in PBS buffer were per-formed at $\lambda = 240$ nm from acidic to alkaline pH values, by successive addition of small aliquots of sodium hydroxide. All absorbance measurements were performed at $\lambda = 240$ nm (refs 54,55) at a final concentration of 250 μM with 1 mM of tris(2-carboxyethyl)phosphine to prevent dimerization in PBS buffer. The basal absorbance of tris(2-carboxyethyl)phosphine at $\lambda = 240$ nm was subtracted from the total absorbance signal in the p$K_a$ determination procedure. All samples were homogenized by strongly stirring immediately before each measurement and their pH value was measured with a Mettler Toledo pH-metre. The p$K_a$ values were determined by fitting the Henderson–Hasselbach equation to the absorbance-pH data.

**Mass spectrometry of L-cysteine and L-homocystine.** ESI-MS experiments were performed with a Thermo LTQ XL linear ion Trap mass spectrometer with electron transfer dissociation source. The electrospray voltage was kept at 4.00 kV, the capillary temperature was set at 350 °C and the capillary voltage was 19 V. A full MS ($m/z$ 107–400) scan was acquired with a nominal mass resolution. Samples were delivered by means of flow injection. MS data was analysed with Xcalibur version 2.0.7 software. L-homocysteine was freshly prepared right before the MS experiment at a pH of 9.2 and a final concentration of 63 μM, and was confirmed by following the $m/z$ molecular ion peak at 136 $m/z$. Dimerization to L-homocysteine was induced by preparing a 63 μM L-homocysteine solution at pH 9.2, and mea-sured after 15 h of sample preparation. L-homocystine was confirmed by following the $m/z$ molecular ion peak at 269 $m/z$. Both species present in the same spectrum were obtained by bubbling the measuring solution with $N_2$ prior to MS determination.

**Data availability.** The MS proteomics data have been deposited to the ProteomeXchange Consortium via the PRIDE partner repository with the data set identifier PXD005981. Additional information and the data supporting this research, including the single-molecule nanomechanics experiments, can be obtained from the corresponding author upon reasonable request.

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

## Acknowledgements

We thank Ainhoa Lezamiz and Dr Palma Rico for protein purification. A.E.M.B. is funded by an EPSRC DTP fellowship. M.M. is funded by a Fight for Sight PhD studentship. G.S. acknowledges support from CNRS through a PICS allocation (PICS07571). This work was supported by the Marie Curie CIG (293462) grant, BBSRC (J00992X/1) grant, Royal Society Research grant (RG120038), BHF grant (PG/13/50/30426) and EPSRC Fellowship (K00641X/1), all to S.G.-M.

## Author contributions

S.G.-M. conceived research. A.E.M.B. and S.G.-M. designed experiments. A.E.M.B. conducted single-molecule mechanical experiments and analysed data. M.M. performed absorbance experiments and small compound mass spectrometry experiments. S.L. performed protein-based mass spectrometry experiments and analysed data.; G.S. performed and analysed DFT calculations. A.E.M.B. and S.G.-M. wrote the paper. All authors contributed in revising and editing the manuscript.

## Additional information

**Competing interests:** The authors declare no competing financial interests.

**Publisher's note**: 

