## [Peer review file · Nature Communications]

Reviewers' comments:

Reviewer #1 (Remarks to the Author):

The submitted manuscript presents a solid and well detailed description of the effect of small molecular weight thiols on the oxidative folding equilibria of elastomeric proteins. Use is made of a polyprotein with eight identical domains whereby disulfide bonds can be broken/formed, changing the mechanic properties. By a combination of atomic force microscopy, DFT calculations and mass spectrometry, the authors are able to confirm that varying the thiols in solution, the equilibria for oxidative folding can be strongly influenced, and this on the basis of very small chemical changes. They are able to link this phenomenon to the reactive and nucleophilic properties of the thiols under study. Overall, I deem this to be a very important contribution to the field, wonderfully supported by the combination of methods chosen. The data and the approaches used are generally well documented. I only have minor comments/requests to place:

1- This is a minor, but at the same time grave point. In Figure 3a, a chemical reaction is presented where the number of atoms and charges do not coincide comparing the left and right hand sides. There is a proton missing, which should probably be included as $-H^+$ under the arrow.

2- In Figure 6c, a pKa of 9.5 is given, contradicting the value in the text and the Supplementary Material. It should read 9.6.

3- The DFT calculations carried out are far from state-of-the-art, but are adequate enough to support the authors' findings. However, I do have to disagree with some particular choices made:

a) in Table 1, the reaction free energies are given for model reactions. A subscript "prot" is used to signal that the molecular species should be considered protein bound. However, in the calculations carried out, a Cys_prot is exactly the same as a Cys. I can understand the use of such a nomenclature in Figure 4, but am strongly against its use in Table 1. The Table refers to computed values, there is no such thing as a Cys_prot in the calculations. Reaction 1 can only have a free energy of 0 (by the way, it should read "0.0" in the table to keep with the number of decimal places in the other cases), since the reactants are equal to the products.

b) the use of Kohn-Sham orbital energies is ill-advised, as well as Mulliken charges (which are extremely sensitive to the choice of basis set). DFT orbitals energies are known to correlate well with computed Hartree-Fock values (Hoffmann and others have pointed this out a long time ago and influenced a whole generation of chemists). The latter have a physical meaning on the basis of Koopmans' theorem. Kohn-Sham orbitals do not. One can accept their use on the basis of experience, but there are better ways to measure nucleophilicity, e.g., through the use of Green's functions. The ionization potential is an observable, HOMO energies are not. If there were two sulfur atoms in the thiols used, the higher-lying orbitals would mix and it would be impossible to draw any conclusions. This being a journal with a very high impact, I would usually recommend the use of better indicators (in the case of Mulliken, something like NPA or IBO-based methods should be used as a replacement). But given that the conclusions will probably be very similar, I would instead recommend moving Figure 5c and d to the Supplementary Information, together with

the NAC results. This way, one reduces the visibility of these poor choices and avoids giving a bad example to other authors.

4- the signalling of steps in the AFM (asterisks and squares) is not made consistently throughout the figures.

Reviewer #2 (Remarks to the Author):

The paper by Beedle et al describes the relation between the chemical reactivity of small Cys-based nucleophiles and their effect on the mechanical folding and elasticity of titin modules. The authors use force-clamp spectroscopy to investigate how reduction of SS bond under force by small molecules containing cysteines such as cysteinylglycine, homocysteine or glutathione, alter the folding behavior of titin models by affecting the life-time of the disulfide intermediates. While S-cysteinylation promotes the successful folding for the titin molecules with their disulfide bond oxidized, S-homocysteinylation renders reduced domains upon folding. Further DFT calculations provide a thermodynamic explanation based on the symmetry of the [S-S-S] intermediate state. Although the paper mainly stresses the chemical importance of these findings, it is noteworthy to mention the potential biological significance of the results since both, L-cysteine and homocysteine are biologically occurring molecules. Overall, I find the work interesting and intriguing as it provides new information on the role of disulfide bond reduction/oxidation on protein folding. Perhaps the most important point is that the authors convincingly provide evidence of a novel mechanism for controlling protein nanomechanics through sound chemical reactivity.

There are a few important points that the authors should consider:

1) Although the authors study cys-based compounds, there are other nucleophiles that could trigger disulfide bond reduction using different mechanisms. In their discussion, they should comment on how their method could be extended in the future to explore other chemical mechanisms to modify protein mechanics, perhaps using other nucleophiles/chemical reagents.

2) Regarding the pKa determination experiments, the authors demonstrated that homocysteine undergoes dimerization by using TCEP in the measuring solution. As a control experiment, the authors should also add TCEP onto the cysteine and NAC pKa determination to rule out the possibility that they also create dimers.

3) Although briefly mentioned in the text, the authors should maybe emphasize that in their MS experiments the degree of chemical modification may change from that obtained in the AFM, since the degree of exposure might be modified by the force. While the trend is definitely correct, perhaps the yield may change.

Reviewer #3 (Remarks to the Author):

Single molecule force spectroscopy has been extensively used to probe the effects of mutations and other structural perturbations on protein mechanics and protein folding. This technique gives access to exquisite detail on the reaction pathways the proteins take and their influence on the mechanical resilience and mechanical function. In this work, the authors open a new set of questions to this type of experimentation, to reveal chemical schemes that modify protein structure and function in biologically important ways. In particular, the authors use low molecular weight thiols to test their reactivity with disulfide bonds in mechanically stable proteins and they uncover a rich interplay between the mechanical strength of the molecule and the chemical reactions that take place. In particular, they discover that the presence of intermediates, and their lifetimes, influences the folding fate of the protein in rather dramatic ways.

For example, even in the absence of any oxidizing agents, the protein is able to successfully refold and reform the disulfide bond. This is because the L-cysteine serves as a nucleophile to initially cleave the disulfide bond, but then the disulfide reforms once the force is removed, either by the attack of the native cysteine or another free thiol. This experiment shows the success of non-enzymatic oxidative folding. In other words, the protein is never under oxidizing conditions, and yet the S_N2 nucleophilic attack presents an alternative pathway. This reaction leaves one sulphur hanging, and it can be re-attacked by the free thiol or by the cysteine.

The details of the nucleophile's structure matter because the presence of an extra methylene group on an analogous homocysteine basically prevented the formation of the disulfide bond. The fact that this nucleophile prevented oxidative folding altogether, but did allow for the rest of the protein to fold without the disulfide resulted in a protein conformation that is much weaker than the one with the disulfide. In that sense, the same protein under attack by two different nucleophiles results in a stiff and in a weak configuration, altering the mechanical strength of the molecule.

What is it that makes the energy of the disulfide between these constructs so different? It could be thermodynamic stability, or kinetic accessibility. This is an important question because it gives insight into the importance of posttranslational modification pathways in the heart muscle, for example. Using DFT calculations, they find that the difference is thermodynamic, and that the pKa, or the equilibrium constant, is responsible for the different behavior of both nucleophiles. This explains why the symmetric L-cysteine is more likely to fold than the other homocysteine, in which there is an energy penalty to get the disulfide to form.

In summary, cysteine is a worse nucleophile than homocysteine and the authors show using DFT simulations that this is because of the distribution of charge inside the molecule. The more charge on the sulphur there is, the more reactive the species. Indeed, this observation is confirmed by a third nucleophile. There, the presence of an acetyl group, which pulls electrons away from the sulphur, folds much better than cysteine, but the disulfide is never reformed. Indeed, the authors show using mass spectrometry that the mixed disulfide is

present throughout.

In order to confirm the crucial role of the mixed disulfide and the folding outcome, another reagent results in a S-P bond in the protein construct. In this case, there is no nucleophilic attack by the native cysteine, and therefore no oxidative folding.

All in all, this constitutes very exciting research because small thiols are present in the plasma and are sure to affect protein reactivity and mechanical stability in vivo. Lessons learned in this study can now be used to design experiments in biological systems to test the importance of these specific pathways.

I believe that this work should be published in Nature Communications due to its wide appeal, and I think that the authors will improve the manuscript if they address the following questions:

1. The authors should stress more that the calculations are not done in the core of the protein, but on isolated small parts. We do not know whether the results would persist in the core of the protein.
2. Why did the authors choose 8 seconds for the waiting time for folding? How does this compare to the folding rate and which percentage would one expect to see folded in this time window?
3. Is there a way in which the authors could speculate on how this mechanism of altering the mechanical stability might play a functional role in biology?
4. Would the results persist on other molecules with disulfides? Does it only work for proteins whose cysteines are in close proximity?
5. How do the authors know that the thiol exchange is always SN2, while phosphorus is thought to undergo SN1?

We thank the three reviewers for their insightful comments, which have truly improved the quality of our paper. Below, please find a point-point, detailed response to the issues raised by the reviewers (*verbatim* in italics).

Response to Reviewer #1

1. -*This is a minor, but at the same time grave point. In Figure 3a, a chemical reaction is presented where the number of atoms and charges do not coincide comparing the left and right hand sides. There is a proton missing, which should probably be included as -H+ under the arrow.*

R: We thank the reviewer for pointing this out. We have now amended Figure 3a accordingly.

2.- *In Figure 6c, a pKa of 9.5 is given, contradicting the value in the text and the Supplementary Material. It should read 9.6.*

R. Again, we thank the reviewer for raising this point. We have now corrected this in Figure 6.

3.- *The DFT calculations carried out are far from state-of-the-art, but are adequate enough to support the authors' findings. However, I do have to disagree with some particular choices made:*

a) in Table 1, the reaction free energies are given for model reactions. A subscript "prot" is used to signal that the molecular species should be considered protein bound. However, in the calculations carried out, a Cys_prot is exactly the same as a Cys. I can understand the use of such a nomenclature in Figure 4, but am strongly against its use in Table 1. The Table refers to computed values, there is no such thing as a Cys_prot in the calculations. Reaction 1 can only have a free energy of 0 (by the way, it should read "0.0" in the table to keep with the number of decimal places in the other cases), since the reactants are equal to the products.

R. The reviewer is correct. Following his/her advice, we have changed this nomenclature in Table 1.

b) the use of Kohn-Sham orbital energies is ill-advised, as well as Mulliken charges (which are extremely sensitive to the choice of basis set). DFT orbitals energies are known to correlate well with computed Hartree-Fock values (Hoffmann and others have pointed this out a long time ago and influenced a whole generation of chemists). The latter have a physical meaning on the basis of Koopmans' theorem. Kohn-Sham orbitals do not. One can accept their use on the basis of experience, but there are better ways to measure nucleophilicity, e.g., through the use of Green's functions. The ionization potential is an observable, HOMO energies are not. If there were two sulfur atoms in the thiols used, the higher-lying orbitals would mix and it would be impossible to draw any conclusions. This being a journal with a very high impact, I would usually recommend the use of better indicators (in the case of Mulliken, something like NPA or IBO-based methods should be used as a replacement). But given that the conclusions will probably be very similar, I would instead recommend

moving Figure 5c and d to the Supplementary Information, together with the NAC results. This way, ones reduces the visibility of these poor choices and avoids giving a bad example to other authors.

R: We really thank the reviewer for these insightful comments, which we definitely take on board.

We agree with the reviewer in that there are much better ways to estimate nucleophilicity, especially using TD-DFT approaches, which would be computationally more expensive and beyond the scope of the current work. Nevertheless, we would like to stress that the estimated reaction free-energies displayed in Table 1 do not suffer from the criticism raised from this reviewer, as they do not depend on the specific orbital energies but rather refer to the total energy of the studied molecules. Therefore, we only use the HOMO energies and the atomic charges as qualitative confirmation of the observed trends in the model reaction free-energies.

Following the reviewer's suggestion, we have now used a more accurate and trustworthy method to estimate the atomic charges by performing a Natural Bond Orbital Analysis (NBO) on the optimized structure in implicit solvent. All the values mentioned in the manuscript and in the Supporting Information Section refer now to these new values. Interestingly, while the exact values slightly differ from those obtained with the previous Mulliken approach (by ~0.2 atomic charge unit), the trend and the quantitative differences between the 3 molecular systems are very similar in both approaches, suggesting that our conclusions are robust.

Following the reviewer's suggestions, we have now moved Fig. 5c and 5d to the SI section, and we have removed the explicit quantification of the HOMO levels in the main text and left it only in the Supplementary Fig. 5. We have now added the values of the computed charges using the NBO model in the main text, on page 8, to read: "Similarly, the effective electrostatic negative charge localised on the attacking sulfur atom in homocysteine is more negative (-0.81 e) than in L-cysteine (-0.77 e)." We once again thank this reviewer for his/her insightful comments.

4.- The signalling of steps in the AFM (asterisks and squares) is not made consistently throughout the figures

R. We thank the reviewer for this comment. Originally, we had used asterisks to represent disulfide reformation (10 nm steps) and squares to represent the presence of a reduced protein (25 nm steps). We now realise that this might not be clear and have opted to change all folding events to asterisks, distinguishing the events by colour instead, where *blue* represents disulfide reformation and *purple* represents reduced protein folding throughout all the figures.

Response to Reviewer #2

1.- Although the authors study cys-based compounds, there are other nucleophiles that could trigger disulfide bond reduction using different mechanisms. In their discussion, they should comment on how their method could be extended in the future to explore

other chemical mechanisms to modify protein mechanics, perhaps using other nucleophiles/chemical reagents.

R: In this study, we have focused on the use of small plasma thiols to disrupt the a protein disulfide bond, occurring through a nucleophilic S_N2 attack. The disruption of the protein disulfide with an S-thiolate gives rise to the presence of a mixed disulfide, the life-time of which regulates protein elasticity. To show the crucial role of the mixed disulfide to eventually reform the disulfide bond—giving rise to a stiff, oxidized protein—, we compared the results obtained with the thiols cysteine (Fig.1), homocysteine (Fig.2) and NAC (Fig 4) with a P-based nucleophile (TCEP, Fig. 3). In contrast to the small thiols, reducing the protein disulfide with TCEP gives rise to an unstable S-P intermediate adduct that can be easily cleaved by solvent molecules. The absence of the mixed disulfide in this case results in two free cysteines, which enable the correct reformation of the protein, albeit lacking the mechanically stiff disulfide bond. Other non-S-based nucleophiles that can break protein disulfides include for example OH-, giving rise to a sulfenic acid moiety, which can rapidly condense with a neighboring thiolate to reform a disulfide bond (Beedle et al., *Nat Comm*, 2016). In theory, selenium-based disulfides should also be able to reduce disulfide bond, and we would like to test this in the future. We do not know how the reactivity of the Se-S mixed adduct will be able to regulate protein elasticity, but it is an exciting and worth-pursuing experiment. Furthermore, the disulfide bond can also be broken through a redox pathway (as opposed to nuclear substitution) with the presence of metals such as zinc (in their (0) oxidation state) in solution (Perez-Jimenez et al., *NSMB*, 2009). In this case, we predict that after disulfide bond rupture, the protein will exhibit two free reduced cysteines that will not be able to reform the disulfide bond. Finally, an exciting way to disrupt the disulfide bond is in the presence of UV light (Soorkia et al., *J. Phys. Chem. Lett.* 2014) although these experiments are challenging when performed in solution due to water absorption. In these experiments, it might be interesting to see how the light-generated radicals would affect the folding dynamics and ultimately the protein's elasticity. Our laboratory is indeed currently working on the exploration of all these new ways to induce disulfide bond rupture and we hope to have positive results in the near future. According to the reviewers's suggestion, we have now spelled out the general potentiality of the method in the discussion section of the revised version of the manuscript, on page 11, to read: "*More generally, our proof-of-principle approach demonstrates the novel use of highly localized chemical reactivity—which can be potentially expanded to other distinct chemical mechanisms—to rationally tailor protein elasticity*".

2.- *Regarding the pKa determination experiments, the authors demonstrated that homocysteine undergoes dimerization by using TCEP in the measuring solution. As a control experiment, the authors should also add TCEP onto the cysteine and NAC pKa determination to rule out the possibility that they also create dimers.*

R: Following the reviewer's advice, we have now conducted this set of

new experiments, whereby we have measured the pK_a of cysteine and NAC in the presence of 1 mM TCEP (see below, Fig R1). Our results demonstrate that the presence of TCEP does not affect the absorbance (and hence the pK_a determination) of neither cysteine or NAC, further suggesting that these two compounds, in contrast to homocysteine, do not dimerize under the experimental conditions. We have now included these measurements in the Supplementary Figure 4, and modified the Methods section of the main text accordingly.

R1. The pK_a determination of cysteine and NAC through absorbance measurements is independent of the presence of TCEP, demonstrating that neither of them dimerize under experimental conditions.

3.- Although briefly mentioned in the text, the authors should maybe emphasize that in their MS experiments the degree of chemical modification may change from that obtained in the AFM, since the degree of exposure might be modified by the force. While the trend is definitely correct, perhaps the yield may change.

R. Thank you. We have further stressed this point in the revised version of the manuscript, on page 6, to read: "Noteworthy, given the different degree of solvent exposure of the protein disulfide bond obtained in MS and mechanical measurements, the yield of protein modification is not comparable in both experimental approaches".

Response to Reviewer #3

All in all, this constitutes very exciting research because small thiols are present in the plasma and are sure to affect protein reactivity and mechanical stability in vivo. Lessons learned in this study can now be used to design experiments in biological systems to test the importance of these specific pathways. I believe that this work should be published in Nature Communications due to its wide appeal, and I think that the authors will improve the manuscript if they address the following questions

R: We thank the reviewer for these positive comments. The reviewer

precisely summarizes our main findings and the scope of our work.

1.- *The authors should stress more that the calculations are not done in the core of the protein, but on isolated small parts. We do not know whether the results would persist in the core of the protein.*

R: **The reviewer is correct. We have further re-stressed this point in the revised version of the paper, on page 6, to read:** *“in this simplified approach any free-energy contributions arising from a protein conformational change upon formation of a native or mixed disulfide are neglected (only the free-energy associated with an isolated disulfide bond rupture is computed)”*.

2.- *Why did the authors choose 8 seconds for the waiting time for folding? How does this compare to the folding rate and which percentage would one expect to see folded in this time window?*

R: **The reviewer raises an interesting point. In our experimental design, we chose a waiting time that is long enough to capture the ‘final’ folded state of the protein exposed to a given perturbation. Being a kinetic process, the folding probability is often described to follow a time-dependency that follows a single exponential. From published (Carrion-Vazquez et al., PNAS 1999) and unpublished data (Fig. R2) we know that after 8 seconds the wt-I27 protein exhibits a ~98% folding yield. Similarly, the probability of disulfide reformation for the I27₂₄₋₅₅ mutant exhibits a ~94% of efficiency after $t_q = 8s$ (Beedle et al., *Nat Comm* 2016). Altogether, the choice of a long folding time ensured that, in our experiments, we could independently capture the folding process and decouple it from the disulfide bond reformation process. We have commented on this point in the revised version of the manuscript, on page 5, to read: “ $t_q = 8s$ (a period of time that is long enough to ensure quantitative protein refolding) triggered...”**

[figure redacted]

3.- *Is there a way in which the authors could speculate on how this mechanism of altering the mechanical stability might play a functional role in biology?*

R. The presence of disulfide bonds within a protein is a natural strategy to regulate protein elasticity employed by a variety of proteins with mechanical function. In general, a reduced disulfide bond allows complete unfolding of the protein. By contrast, under oxidising conditions the disulfide bond creates a covalent staple that shortcuts the protein, thus impeding complete mechanical unfolding. For example, the vascular cell adhesion molecule-1 (VCAM-1) is composed of seven Ig domains harbouring a buried disulfide bond in their core, covalently stabilizing about half of the module against unravelling (Vogel et al., *Annu. Rev. Biophys. Biomol. Struct.*, 2006). Moreover, the crystal structure of the Ig1 domain of titin reveals a disulfide bond between Cys36 and Cys61 that massively restricts the extension of the protein under force (Mayans et al., *Structure* 2001; Li et al., *JMB*, 2003). Other Ig modules, including ICAM-1, ICAM-2 and human CD2 have also a disulfide bond at or near the N- or C-terminal loops, which limit the extent to which these modules can be unravelled by force (Carl et al., *PNAS*, 2001). These are only a few examples in biology whereby the mechanical stability of a protein can be dramatically altered by the reduction/reformation of an individual disulfide bond. More specifically, S-thiolation resulting from the attack of a protein disulfide bond by a small thiol induces a mixed disulfide conformation, the life-time of which can regulate the mechanics of the protein. For example, it is enticing to speculate that different buried cysteines in different Ig domains of titin (Alegre-Cebollada, *Cell*, 2014), or in different non-structured domains such as the N2Bus region (Grützner et al., *BJ*, 2009) can be post-translationally modified when exposed to the solution through mechanical unfolding by the small plasma thiols here studied (such as cysteine or homocysteine). Upon force withdrawal, the different PTModified proteins could eventually reform the disulfide bond. Hence, it is possible that these naturally-occurring small thiols serve as a natural strategy to modify tissue elasticity within biologically-relevant timescales. In the near future, we will test whether the single molecule observations can be recapitulated in the cellular (muscular) context.

4.- *Would the results persist on other molecules with disulfides? Does it only work for proteins whose cysteines are in close proximity?*

R: Our experiments are conducted on a model protein that allows us to independently decouple mechanical folding from disulfide bond reformation. Given the dramatically distinct reactivity of the induced mixed disulfide bond triggered by each independent thiol nucleophile, we believe that our findings could be made general to any protein harbouring two cysteines that are close enough in the 3D structure to reform a disulfide bond.

5.- *How do the authors know that the thiol exchange is always SN2, while phosphorus is thought to undergo SN1?*

R: The thiol-disulfide interchange has been long shown by a number of studies to proceed through a simple S_N2 type nucleophilic substitution mechanism. The S_N2 mechanism is consistent with a one-step reaction via a single transition state complex with no intermediate formation. The

nucleophile is the deprotonated thiolate anion, which attacks the reacting sulfur of the disulfide moiety (P. Nagy, *Antioxidants & Redox Signaling*, 2013). Several pieces of theoretical and experimental evidence suggest a linear trisulfide-like transition state, with the negative charge being delocalized, but most abundant on the attacking and leaving sulfurs (Bach et al., *J Org Chem*, 2012; Fava et al., *JACS*, 1957; Fernandes and Ramos, *Chem Eur J*, 2004; Liang et al, *ACS Nano*, 2009, Singh and Whitesides, "The chemistry of sulphur-containing functional groups", 1993, Wiley; Whitesides et al., *J Org Chem*, 1983).

Disulfide bond rupture induced by a phosphorous-containing nucleophile also occurs initially through an S_N2 mechanism. This step implies the attack of the phosphorous atom along the S-S bond axis displacing a thiolate anion attending the formation of a thioalkoxy-phosphonium cation. In a second subsequent step, the cationic intermediate undergoes hydrolysis to form a phosphine oxide and a second equivalent of thiolate anion (Dmitrenko et al., *J. Org. Chem.* 2007; Burns et al., *J. Org. Chem.* 1991). This last hydrolysis step, occurring in protic solvents, can be classified as a S_N1 reaction (Fig R3). The high reactivity of this S-P intermediate results in two free, reduced cysteines that can correctly fold into a final protein structure devoid of the disulfide bond. We have now clarified this point in the revised version of the manuscript, on page 6, to read: "In the case of the P-based nucleophile TCEP, the force-induced S_N2 disulfide cleavage⁴⁸ obviously does not yield a stable mixed disulfide, but rather a highly reactive mixed S-P thioalkoxyphosphonium cation intermediate⁴⁹ (Supplementary Fig. 3), which can be easily displaced by water molecules to render a completely unfolded protein with two reduced thiolates (Fig. 3a)" and we have included the figure in the supporting information section (new Supplementary Fig. 3).

R3. TCEP induces disulfide bond reduction through an S_N2 chemical reaction to create an unstable S-P thioalkoxyphosphonium cation that can be rapidly attacked by water molecules, resulting in two reduced thiolates.

REVIEWERS' COMMENTS:

Reviewer #1 (Remarks to the Author):

I am pleased with the changes the authors carried out and strongly recommend the publication of the manuscript as is.

Reviewer #2 (Remarks to the Author):

I think the authors have carefully and thoughtfully addressed the comments and requests of the 3 reviewers. This revised manuscript is suitable for publication.

Reviewer #3 (Remarks to the Author):

The authors have satisfied all my concerns and adequately addressed the issues raised by the reviewers. I recommend the article for publication as is.